# Surface Characterization and Electrical Properties of Low Energy Irradiated PANI/PbS Polymeric Nanocomposite Materials

**Numa A. Althubiti [1], Nuha Al-Harbi [2,*], Rabab K. Sendi [2], Ali Atta [1,*] and Ahmed. M. A. Henaish [3,4]**

1   Physics Department, College of Science, Jouf University, P.O. Box 2014, Sakaka, Saudi Arabia
2   Departments of Physics, Faculty of Applied Sciences, Umm Al-Qura University, Makkah 21955, Saudi Arabia
3   NANOTECH Center, Ural Federal University, Ekaterinburg 620002, Russia
4   Physics Department, Faculty of Science, Tanta University, Tanta 31527, Egypt
*   Correspondence: nfhariby@uqu.edu.sa (N.A.-H.); aamahmad@ju.edu.sa (A.A.)

**Abstract:** In this work, nanocomposite samples of polyaniline (PANI) and lead sulfide nanoparticles (PbSNPs) were prepared, utilizing the solution preparation method, for implantation in energy storage elements. The PANI/PbS films were irradiated by different fluences of oxygen beam: $5 \times 10^{16}$, $10 \times 10^{16}$, and $15 \times 10^{16}$ ions.cm$^{-2}$. The composite was investigated by XRD, SEM, DSC, and FTIR. After ion irradiation, the $T_g$ and $T_m$ values decreased by 4.8 °C and 10.1 °C, respectively. The conductivities, electrical impedances, and electrical moduli of untreated and irradiated samples were examined in frequencies ranging from $10^2$ Hz to 5 MHz. Moreover, the ion beam caused a modification in the dielectric characteristics of PANI/PbS. The dielectric constant $\varepsilon'$ was improved from 31 to 611, and the electrical conductivity increased from $1.45 \times 10^{-3}$ S/cm to $25.9 \times 10^{-3}$ S/cm by enhancing the fluence to $15 \times 10^{16}$ ions.cm$^{-2}$. Additionally, the potential energy barrier, $W_m$, decreased from 0.43 eV to 0.23 eV. The induced changes in the dielectric properties and structural characteristics of the PANI/PbS samples were determined. These modifications provide an opportunity to use irradiated PANI/PbS samples for several applications, including microelectronics, batteries, and storage of electrical energy.

**Keywords:** polymeric composites; ion irradiation; dielectric characteristics; energy applications

## 1. Introduction

Polymer nanocomposite materials have recently received much interest in developing their desirable characteristics for energy storage applications [1,2]. They are being developed due to their properties of structural patterns, mechanical performance, and electric characteristics [3]. Conducting polymer composites are gaining popularity in electrical devices [4,5]. Furthermore, PANI is an excellent conducting polymer, hence its use in sensors, solar cells, and catalytic activity [6,7]. However, PANI has some limitations due to its physico-chemical characteristics and low solubility [8], thus it is important to mix it with certain materials to overcome these difficulties and increase its utilization in the energy industry [9]. Metal particle inclusion on the surfaces of PANI has been described in the literature. Alotaibi et al. used a casting solution technique to create a polyaniline/lead sulfide polymer nanocomposite for use in energy applications [10]. For supercapacitors manufacturing, Atta et al. used polyaniline/silver oxide/silver composite electrodes. They found that the composites are promising electrodes with excellent capacitance effectiveness [11].

The incorporation of inorganic nanofillers into a polymeric matrix leads to remarkably higher dielectric properties [12,13]. Dispersion of nano-size fillers is more desirable than micro-size fillers, because nanoparticle fillers improve both the electrical and mechanical properties of the polymeric material [14]. PbS conductive fillers are used in various applications, such as sensors, solar cells, optoelectronics, and storage devices [15–18]. PbS

can be employed in energy devices because of their conductivity, long life, and stability. The average bandgap of PbS has been recorded as being of the order of 0.4 eV, while the absorption coefficient is of the order of $10^5$ cm$^{-1}$, and the radius of Bohr exciton is nearly 18.5 nm. Furthermore, the glass transition temperature of a PANI/PbS composite is nearly 130 °C, and its melting temperature is roughly 260 °C. Moreover, PANI/PbS has emerged as a potential substance for bio-sensing, and fuel cell technologies [19]. The conductive PANI/PbS composites, on the other hand, are appealing materials for electrical storage applications because of their adaptability and ease of synthesis.

Ion irradiation improves the dielectric response performance of the composites, as well as having other effects, such as the modification of the complex permittivity [20,21]. Furthermore, dielectric analysis is critical in creating electrolyte devices, including sensors, batteries, and fuel cells [22]. In recent years, ion irradiation has been shown to be a promising method for producing long-term changes in the dielectric properties of the polymer matrix [23]. This is because ion irradiation promotes polymer matrix changes, including cross-linking, carbonization, oxidation, and free radical production [24,25]. The novelty of this work is to modify the properties of the manufactured PANI/PbS film using a homemade ion source [26]. The ion source requires little instrumentation and maintenance. Consequently, based on the properties of the source, oxygen beam interactions induce defects, vacancies, and changes in the target's properties. In this work, the effect of the low energy oxygen beam on the structural and dielectric behavior was evaluated for both pure and irradiated samples. The outcomes of this study open the way for the use of irradiated PANI/PbS composites for energy purposes.

## 2. Results and Discussion

### 2.1. SRIM/TRIM Simulation Data

The images in Figure 1a–d were acquired from SRIM and TRIM simulations, which assess damage occurrences quickly, by estimating the ion range, vacancies, and distribution, as shown in Figure 1. They were plotted by applying 3 keV oxygen energy striking the PANI/PbS target at depths of 1000 A°. The vacancy distributions, and ionization, of the PANI/PbS composite were determined by the energy loss from hitting oxygen ions with recoil atoms [27,28]. In composites, the penetrating ions are decelerated by triggering the objective's electronic system and, therefore, by transferring momentum to the target atoms. The intense oxygen hits can induce damage to composite atoms that, notably, induce chain scission. Electronic excitation results in the formation of radicals that can swiftly link polymeric chains [29]. The atom displacements ion pathway of the penetrating oxygen ions is ~151 A°, as shown in Figure 1a. A series of randomly dispersed collisions, incorporating PANI/PbS atoms of depth 1000 Å, is depicted in Figure 1b. The recoiled atoms were granted sufficient energy to depart the structure and interact with additional H, N, and C atoms. Figure 1c depicts the colliding interactions of oxygen ions with the target's vacancy, resulting in target damage [30]. The data depicted in Figure 1c demonstrate that the density of C recoil is approximately $12 \times 10^6$ atoms/cm$^2$, which is higher than the density of H recoil (~$10 \times 10^6$ atoms/cm$^2$) and N recoil (~$2 \times 10^6$ atoms/cm$^2$). On the other hand, Figure 1d shows the ionization induced by recoil atoms and incident ions. Ionization produced from recoiled atoms is nearly the same as that generated by the penetrated oxygen ions (Figure 1d) [31].

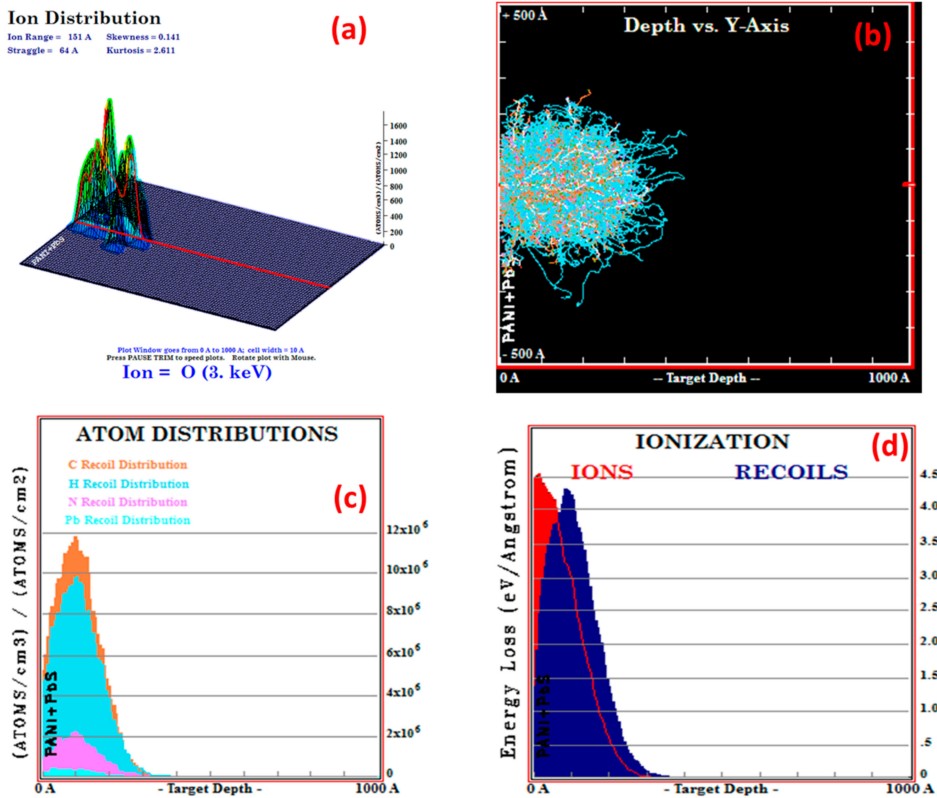

**Figure 1.** (**a**) Ion range of oxygen beam with PANI/PbS, (**b**) distribution of oxygen ions with PANI/PbS, (**c**) target vacancies influenced by collided oxygen ions, and (**d**) ionization of collided oxygen ions as well as recoil atoms.

### 2.2. Structure of the PANI/PbS

The XRD measurements of pure and irradiated PANI/PbS are plotted in Figure 2, which shows some distinct crystalline peaks for PANI/PbS at $2\theta = 26°$, $30°$, $33°$, $43°$, $52°$, $54°$, $62°$, and $66°$ corresponding to (110), (111), (022), (132), (170), (222), (311), and (133), respectively. The obtained results reveal a reduction in the crystalline intensity of the irradiated films that will be verified by FTIR and SEM results. Additionally, the whole width at half maxima from the diffraction peaks increased upon irradiation. This tendency is related to the composite films' decreased crystallization [32].

XRD was employed to evaluate the crystallite sizes, lattice strains, dislocation densities, as well as distortion coefficients. The Debye–Scherrer equation is used to estimate the crystallite size (**D**) of pure and treated PANI/PbS using the following formula [33]:

$$\mathbf{D} = \frac{0.94\lambda}{\beta\,\cos\theta} \tag{1}$$

$\beta$ refers to the entire width of the (111) plane, $\theta$ is the diffraction Bragg angle, and $\lambda$ indicates the X-ray tube wavelength. The diameter (**R**) is calculated by the following equation [34]:

$$\mathbf{R} = \frac{\lambda}{\sin\beta\cos 2\theta} \tag{2}$$

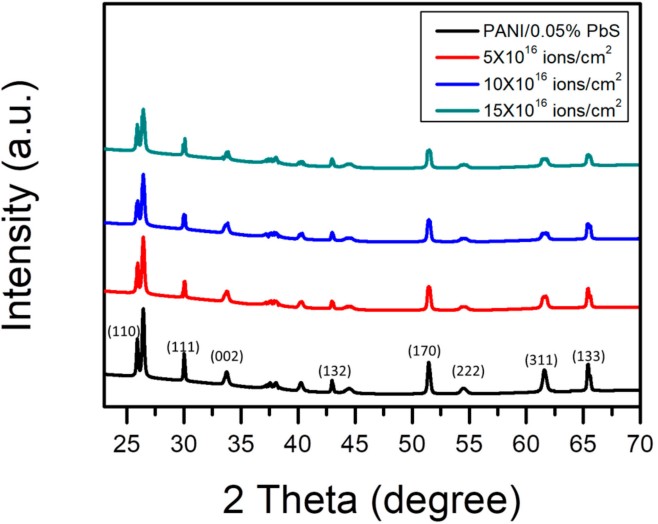

**Figure 2.** XRD of pure and treated PANI/PbS.

The crystallite size of PbS is 45.88 nm for pure PANI/PbS, which is reduced to 40.36 nm after irradiation by $15 \times 10^{16}$ ions/cm$^2$, while R decreased from 3.04 μm to 2.67 μm, as shown in Table 1. These results were a consequence of the creation of a disordered structure in the irradiated samples. The variable dislocation density ($\delta$) was computed using the formula [34]:

$$\delta = \frac{1}{D^2} \tag{3}$$

**Table 1.** The induced micro-structural parameters of the pure and irradiated PANI/PbS.

|  | D [nm] | R [μm] | $\delta$ [$10^{-4}$ Lines/m$^2$] | $\varepsilon$ [$10^{-3}$] | g (%) |
|---|---|---|---|---|---|
| PANI/PbS | 45.88 | 3.04 | 4.75 | 3.44 | 0.0138 |
| $5 \times 10^{16}$ ions/cm$^2$ | 44.23 | 2.93 | 5.11 | 3.56 | 0.0143 |
| $10 \times 10^{16}$ ions/cm$^2$ | 42.18 | 2.79 | 5.62 | 3.74 | 0.0150 |
| $15 \times 10^{16}$ ions/cm$^2$ | 40.36 | 2.67 | 6.13 | 3.92 | 0.0157 |

The dislocation parameter increased from $4.75 \times 10^{-4}$ lines/m$^2$ for PANI/PbS to $6.13 \times 10^{-4}$ lines/m$^2$ after irradiation. The lattice strain ($\varepsilon$) is estimated by [35]:

$$\varepsilon = \frac{\beta}{4\tan\theta} \tag{4}$$

The actual film's lattice strain increased from $3.44 \times 10^{-3}$ for the pure PANI/PbS film to $3.92 \times 10^{-3}$ after irradiation. This change is because of particle size reduction, as well as particle misalignment after ion irradiation. Then, the distorted parameters (**g**) of the untreated and irradiation samples were determined using the following relationship [35]:

$$\mathbf{g} = \frac{\beta}{\tan(\theta)} \tag{5}$$

**g** rises from 0.0138 in the untreated PANI/PbS film to 0.0157 after beam irradiation. This demonstrates that the ion beam induces effects on the structure by modifying the crystalline structure without changing the crystals' alignment.

### 2.3. DSC of PANI/PbS

To demonstrate the thermal properties of the synthesized samples before and after irradiation, a differential scanning calorimeter (DSC) was used. Ion beam irradiation is expected to induce alteration in the polymer chain by scission, inducing a reduction in the crystalline phase, and lowering the transition temperature ($T_g$) and melting temperature ($T_m$) [36], as depicted in Figure 3. Both $T_g$ and $T_m$ values fall after ion irradiation. The $T_m$ decreased by 4.8 °C, and the $T_g$ decreased by 10.1 °C compared to the un-irradiated film. In addition, the results obtained demonstrate a decrease in the crystalline structure of the irradiated film, which could be attributed to polymeric chain scission after irradiation, in line with the FTIR and XRD data.

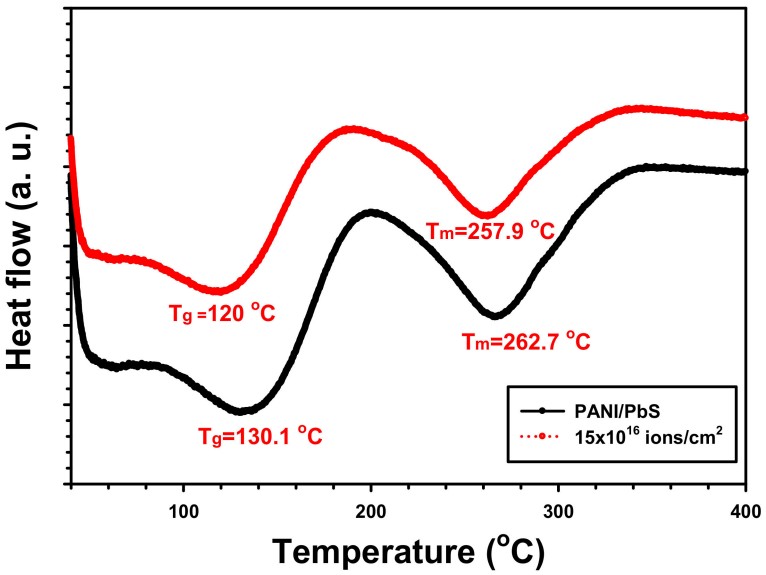

**Figure 3.** DSC thermograms of pure and irradiated PANI/PbS.

### 2.4. FTIR of PANI/PbS

The FTIR of the pure and treated PANI/PbS is recorded in Figure 4. As illustrated, the peak at ~3420 cm$^{-1}$ is assigned to an O−H bending vibration or to N–H asymmetric absorptions [37]. The absorption peak at 900 cm$^{-1}$ is due to the hetero-polar bond of PbS. The bands observed in all the irradiated samples, at 750 cm$^{-1}$ and at 462 cm$^{-1}$, are related to the C−H aromatic ring and the PbS vibration, respectively [38]. These peaks were slightly shifted after irradiation, which was caused by an increase in the energy of interaction between irradiation with PbS/PANI. In addition, the figure demonstrates that the peaks of the irradiated PbS/PANI are lower than those of the untreated film. Furthermore, the shift and intensity reduction of the bands after irradiation suggests chain scission of the irradiated films [39].

### 2.5. Surface Morphology of PANI/PbS

The morphology of pristine and treated PANI/PbS films is imaged in Figure 5a–d. As shown in Figure 5a, the morphology of PANI/PbS is homogeneously distributed, with some agglomerated and self-assembly nano-porous regions [40,41]. Moreover, SEM migrographs of PANI/PbS films treated with different oxygen ion fluence are shown in Figure 5b–d. The images of the irradiated surface show slight changes in the morphology after being treated with oxygen ion fluence. After exposure to radiation, significant changes in the surface morphology were found. The ion irradiance of the PANI/PbS film causes the creation of porous clusters, which enhances the roughness of the films. When the radiation fluence is raised, the surface's ripple becomes more pronounced, and distinct lamellar structures begin to form. A significant level of roughness is produced as a result of the oxygen beam's

ability to break more large chains. This alteration in morphology of irradiated PANI/PbS is primarily responsible for improving the properties of the composite [42].

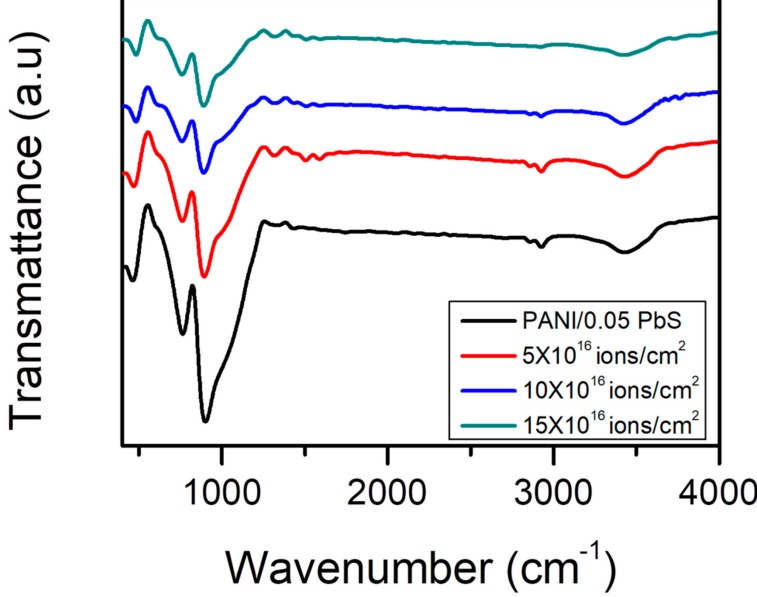

**Figure 4.** FTIR of the untreated and irradiated PANI/PbS samples.

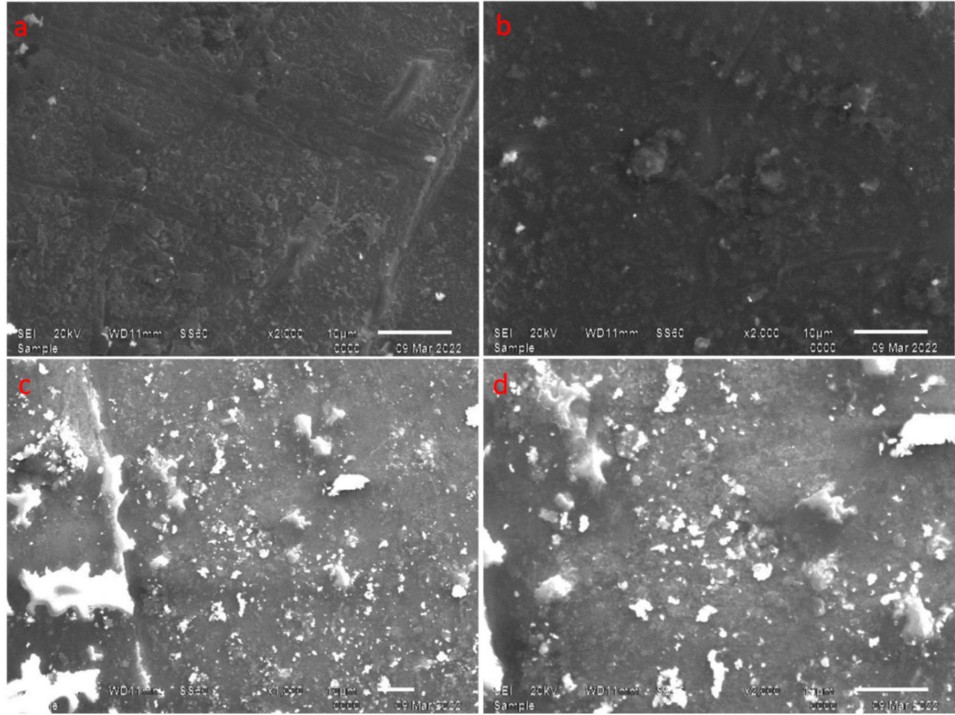

**Figure 5.** SEM image of (**a**) un-treated PANI/PbS and (**b–d**) the films irradiated by $5 \times 10^{16}$, $10 \times 10^{16}$, and $15 \times 10^{16}$ ions/cm$^2$, respectively.

### 2.6. Dielectric Properties of PANI/PbS

The dielectric property analysis is an extremely sensitive tool for revealing relevant information regarding structural behavior. The relationship that gives the dielectric permittivity ($\varepsilon^*$) is [43]:

$$\varepsilon^* = \varepsilon' - i\,\varepsilon'' \tag{6}$$

And the relationship that gives the real $\varepsilon'$ dielectric constant is [44]:

$$\varepsilon' = \frac{c \cdot d}{\varepsilon_o \cdot A} \tag{7}$$

where **c** is the capacitance, **t** is the thickness, and **A** is the area. Figure 6 depicts the variations in $\varepsilon'$ with frequency for pure and treated PANI/PbS films. In the beginning, the decrease in the dielectric value for all films is evident at low frequencies. Furthermore, by increasing the frequency, the $\varepsilon'$ has a virtual value constant, which might be because dipoles have little opportunity to orient themselves [45]. Following irradiation, $\varepsilon'$ increased, this was caused by the production of many defects, and chain scission, in the composite films. Consequently, due to changes in the polarization properties of the irradiated films, these defects were increased through homo-polar linkages between the conduction and valence bands [45]. In addition, the charge transport complexes improve the dielectric characteristics of PANI/PbS films, which contributes to the increase of $\varepsilon'$ for irradiated materials [46]. The $\varepsilon'$ at 100 Hz for PANI/PbS was 31 and it increased up to 341 after $5 \times 10^{16}$ ions·cm$^{-2}$ irradiation, and reached 611 after $15 \times 10^{16}$ ions·cm$^{-2}$ irradiation. The value of $\varepsilon'$ was increased with irradiation fluence due to bond breaking, as investigated with FTIR spectra. As noted previously with regards to the FTIR spectra, there was an overall decrease in peak intensity following irradiation, which could be attributed to the considerable production of unsaturated bonds and the generation of gap states following irradiation [47]. The high dielectric constants of the irradiated films showed their potential for use in energy storage systems.

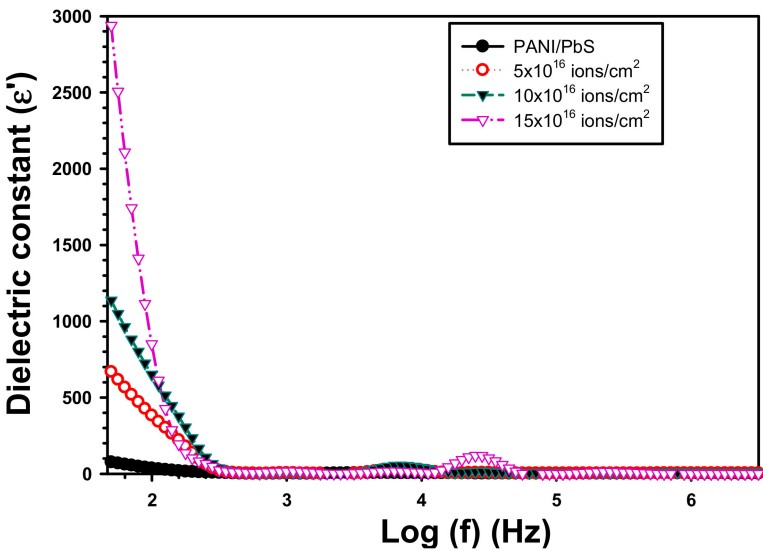

**Figure 6.** Dielectric constant $\varepsilon'$, with frequency for pristine and treated PANI/PbS samples.

The dielectric $\varepsilon''$ is computed using the following relationship [48]:

$$\varepsilon'' = \varepsilon' \tan\delta \tag{8}$$

Figure 7 depicts the change in dielectric loss with frequency for the pristine and treated samples. The $\varepsilon''$ is reduced by increasing the frequency, but a considerable rise in $\varepsilon''$ with increased oxygen fluence is caused by the produced defect. Table 2 shows that the dielectric loss $\varepsilon''$ is improved from 29 for pure PANI/PbS to 65.5 for $5 \times 10^{16}$ ions.cm$^{-2}$ and 126 for $15 \times 10^{16}$ ions.cm$^{-2}$. This rise was caused by interfacial polarization boundaries as a result of irradiation.

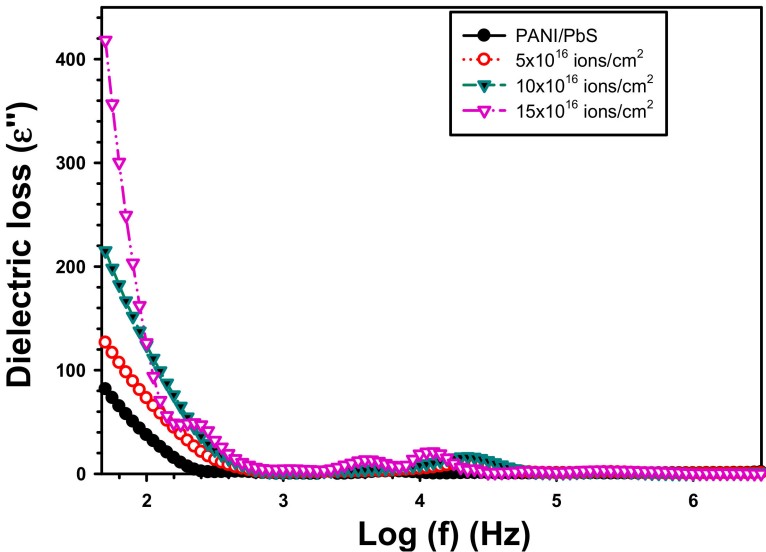

**Figure 7.** Dielectric loss ($\varepsilon''$) with frequency for pristine and treated PANI/PbS samples.

**Table 2.** The $\varepsilon'$, $\mathbf{M'}$, $\mathbf{M''}$, $\varepsilon''$, $\mathbf{U}$ and $\sigma_{ac}$ of pure and treated PANI/PbS at a frequency of 100 Hz.

| | $\varepsilon'$ | $\varepsilon''$ | $\mathbf{M'}$ | $\mathbf{M''}$ | $\sigma_{ac}$ (S/cm) | U(J/m³) |
|---|---|---|---|---|---|---|
| PANI/PbS | 31 | 31.6 | 0.046 | 0.037 | $1.45 \times 10^{-3}$ | $0.17 \times 10^{-3}$ |
| $5 \times 10^{16}$ ions/cm² | 341 | 65.5 | 0.040 | 0.015 | $2.15 \times 10^{-3}$ | $1.5 \times 10^{-3}$ |
| $10 \times 10^{16}$ ions/cm² | 580 | 111 | 0.014 | 0.005 | $2.90 \times 10^{-3}$ | $2.56 \times 10^{-3}$ |
| $15 \times 10^{16}$ ions/cm² | 611 | 126 | 0.004 | 0.0005 | $25.9 \times 10^{-3}$ | $2.7 \times 10^{-3}$ |

The real ($\varepsilon'$), as well as the imaginary ($\varepsilon''$), permittivity coefficients are connected to the accumulated and released energies, respectively [49]. The impact of ion bombardment on the dielectric properties of PVDF doped with BaTiO$_3$ [50] was also reported by Sharma et al. for comparison. The dielectric permittivity was found to be higher in the treated samples when compared to the untreated ones, since the observed change in permittivity can be traced back to the formation of free radicals and scission in the irradiated composite. Therefore, ion radiation exposure causes significant changes in the dielectric properties.

The electrical modulus, $\mathbf{M^*}$, is provided by the following formula [50]:

$$\mathbf{M}^* = \frac{1}{\varepsilon^*} = \mathbf{M}' + \mathbf{i}\,\mathbf{M}'' \tag{9}$$

$\mathbf{M'}$ is real permittivity, $\varepsilon^*$ is complex permittivity, and $\mathbf{M''}$ is imaginary permittivity. The moduli $\mathbf{M'}$ and $\mathbf{M''}$ are provided by the following formula [51]:

$$\mathbf{M}' = \frac{\varepsilon'}{\varepsilon'^2 + \varepsilon''^2} \tag{10}$$

$$\mathbf{M}'' = \varepsilon'' / \left( \varepsilon'^2 + \varepsilon''^2 \right) \tag{11}$$

Figure 8 depicts the relationship of $\mathbf{M'}$ with frequency for pristine and treated PANI/PbS samples irradiated by $5 \times 10^{16}$, $10 \times 10^{16}$, and $15 \times 10^{16}$ ions·cm$^{-2}$. In the low-frequency range, $\mathbf{M'}$ grows exponentially with increasing frequency. As the frequency rises, more dipolar groups are produced, and the dipolar groups contributions decline because of it becomes difficult to orient. [52]. As shown in Table 2, at $10^2$ Hz, the value of $\mathbf{M'}$ decreases from 0.046 for pure PANI/PbS, to 0.040 for $5 \times 10^{16}$ ions·cm$^{-2}$, and to 0.004 for $15 \times 10^{16}$ ions·cm$^{-2}$. The modulus was reduced with irradiation because charge carriers contributed a dipolar

nature and hence changed the localization of the charge density. For comparison, Atta et al. [53] investigated the effect of ion fluence on the dielectric modulus **M′** for PVA/MWCNT. They found at 1 MHz, the **M′** was lowered from 0.050 for the untreated to 0.013 for the irradiated film.

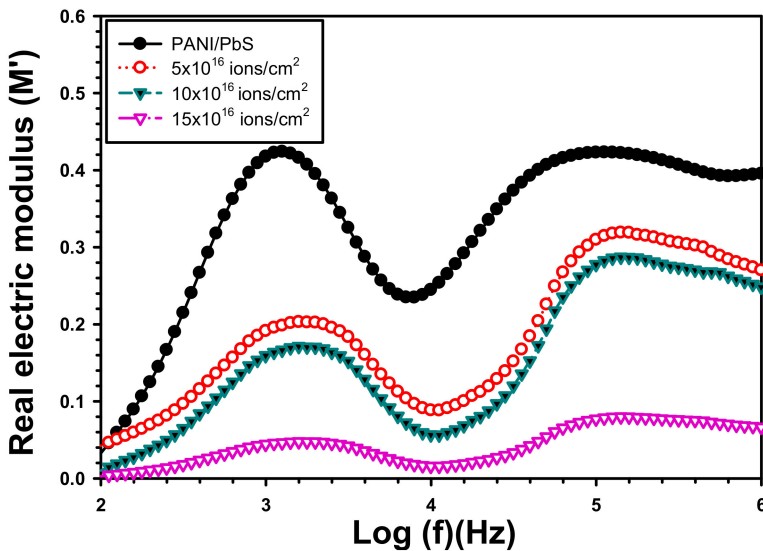

**Figure 8.** Variation of the electrical modulus $\varepsilon'$ with frequency for pristine and treated PANI/PbS samples.

Figure 9 illustrates the change of **M″** by frequency of pristine and treated PANI/PbS films. A peak of **M″** is observed, indicating the existence of a relaxation. On increasing the frequency, dipolar groups became harder to reorient, and the dipolar groups' contribution to the permittivity decreased. Furthermore, at a frequency of 100 Hz, the **M″** lowered from 0.037 for pure PANI/PbS, to 0.015 for $5 \times 10^{16}$ ions·cm$^{-2}$, and to 0.0005 for $15 \times 10^{16}$ ions·cm$^{-2}$. With irradiation, the peak **M″** intensity of pure PANI/PbS shifted to higher frequencies, indicating that the relaxation time ($\tau_r$) was decreased according to the formula [54]:

$$\tau_s = \frac{1}{2\pi f_p} \tag{12}$$

where $f_p$ denotes the frequency at the relevant relaxation peak, and $\tau_s$ denotes the time of relaxation. The $\tau_s$ for pure PANI/PbS was $6.67 \times 10^{-6}$ s, this decreased to $1.249 \times 10^{-6}$ s for $5 \times 10^{16}$ ions.cm$^{-2}$, to $0.92 \times 10^{-6}$ s for $10 \times 10^{16}$ ions·cm$^{-2}$, and reached $0.77 \times 10^{-6}$ s for $15 \times 10^{16}$ ions·cm$^{-2}$. This effect is due to enhanced mobility caused by the oxygen beam, leading to a decrease of $\tau_s$ [55].

Impedance **Z\*** is estimated by the following formula [56]:

$$\mathbf{Z}^* = \mathbf{Z}' + \mathbf{iZ}'' \tag{13}$$

where **Z\***, **Z′**, **Z″** are the complex, real, and imaginary impedances, respectively. As demonstrated in Figure 10, **Z′** reduced with frequency, and became constant at higher frequencies. This is because the free charge carriers induced conduction at low frequencies and more stable impedance at higher ones. The irradiated film behaved similarly to the untreated films. As previously stated, the **Z′** dropped on increased irradiation due to the induced free charge carriers.

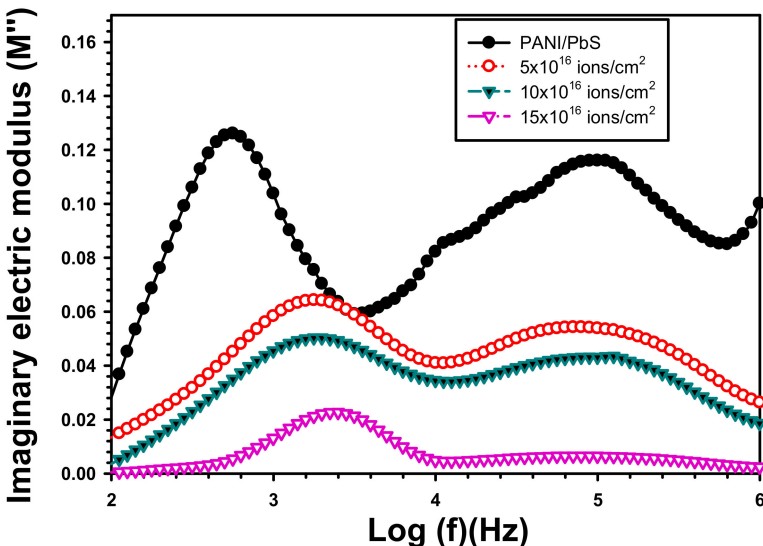

**Figure 9.** Electrical modulus **M″** with frequency for pristine and treated PANI/PbS samples.

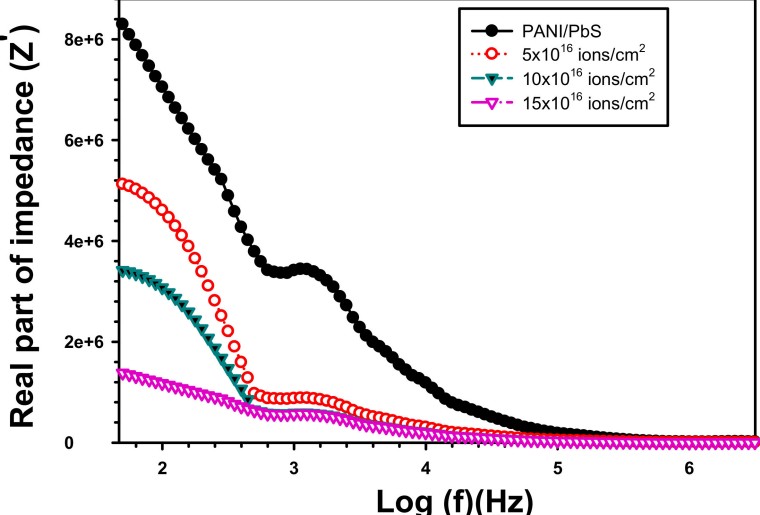

**Figure 10.** Real impedance **Z′** with frequency for pristine and treated PANI/PbS.

Figure 11 depicts the change in **Z″** value with frequency for untreated and treated PANI/PbS films. It is obvious that the behavior of the **Z″** value varies with frequency, just like the behavior of **Z′**. As the conductivity and free charge carriers increase with irradiation, the values of **Z″** gradually decrease [57]. The peaks in both the pure and treated films are because of dielectric relaxation, which proves that the irradiated samples are more suitable for storage systems.

For storage device applications, the energy density (**U**) is determined by [58]:

$$\mathbf{U} = \frac{1}{2}\varepsilon'\varepsilon_o\mathbf{E}^2 \tag{14}$$

where **E** is the field ~1 V/m, and $\varepsilon_o$ is the permittivity, ~0.885 × 10$^{-12}$ C$^2$/N·m$^2$. Figure 12 reveals the influence of frequency on the energy density, U. The un-irradiated PANI/PbS had an energy density of 0.17 × 10$^{-3}$ J/m$^3$, which increased to 1.5 × 10$^{-3}$ J/m$^3$ for 5 × 10$^{16}$ ions·cm$^{-2}$ and reached 2.7 × 10$^{-3}$ J/m$^3$ at 15 × 10$^{16}$ ions·cm$^{-2}$. This is due to the ion beam inducing faster charge transfer in the irradiated films. Moreover, ion irradiation causes different types of polarization modes. The irradiation PbS/PANI interface complicated electric relaxation behaviors and caused changes in the internal electric field,

such as through dipole correlations. Exposure to energetic ions could be utilized to modify the structure and dielectric characteristics of polymeric systems [59]. The energy delivered by the ion beam influences the necessary energy for charge transport. Depending on the ion fluence, ion beam energies, as well as the composition of the polymer matrices, incoming ions can produce scission or cross-linking reactions. This process may result in new defects, radicals, and even polar bonds. This research indicates that irradiated PANI/PbS films have dielectric properties that make them suitable for energy storage.

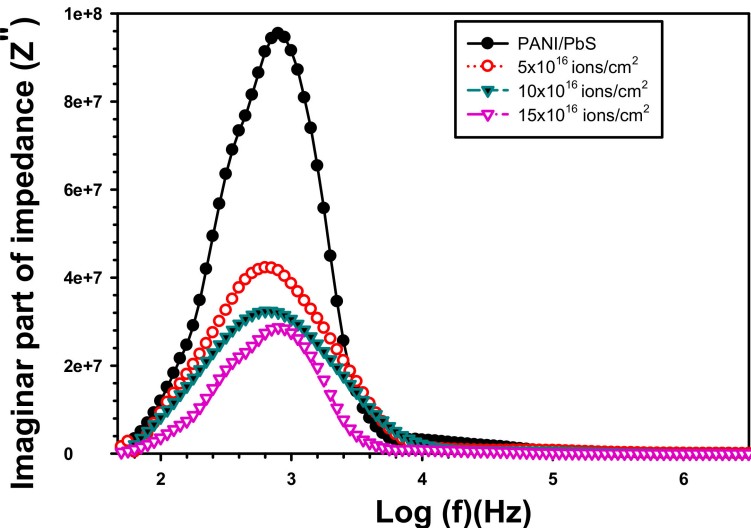

**Figure 11.** Imaginary impedance **Z″** with frequency for pristine and treated PANI/PbS.

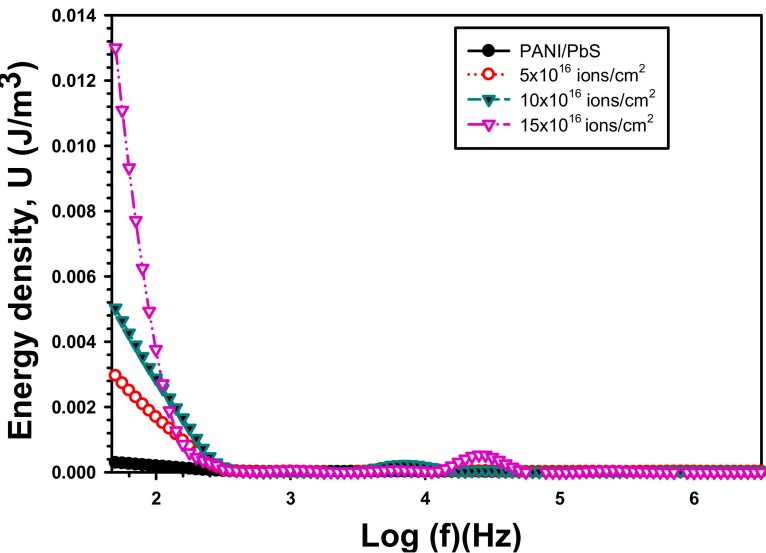

**Figure 12.** Energy density **U** with frequency for pristine and treated PANI/PbS samples.

The **ac** conductivity $\sigma_{ac}$ is determined by [60]:

$$\sigma_{ac} = 2\pi f \varepsilon_o \varepsilon''$$  (15)

where $\varepsilon_o$ is permittivity, **f** is frequency, and $\varepsilon''$ is the loss. Figure 13 demonstrates the change in $\sigma_{ac}$ with frequency. Notably, the conductivity of all films was enhanced by increasing the frequency. Furthermore, at lower frequencies, there is a slight change in ac conductivity; but, at higher frequencies, there is a large shift in conductivity owing to the activation of trapped charges. As noted previously, irradiation enhanced the conductivity

of samples by increasing the number of charge carriers. The $\sigma_{ac}$ at 100 Hz improved from $1.45 \times 10^{-3}$ S/cm for PANI/PbS film, to $2.15 \times 10^{-3}$ S/cm after irradiation with $5 \times 10^{16}$ ions/cm$^2$, and to $25.9 \times 10^{-3}$ S/cm upon irradiation with $15 \times 10^{16}$ ions/cm$^2$. This improvement in $\sigma_{ac}$ was due to polymer scissioning, which led to faster ionic transport across the chains. Abdelhamied et al. investigated the effects of oxygen beam irradiation on the electrical conductivity of PVA/PANI/Ag at a frequency of 100 Hz [61]. They found that after irradiation, the conductivity improved from $1 \times 10^{-8}$ S/cm for the untreated, to $1.8 \times 10^{-7}$ S/cm for the treated composite.

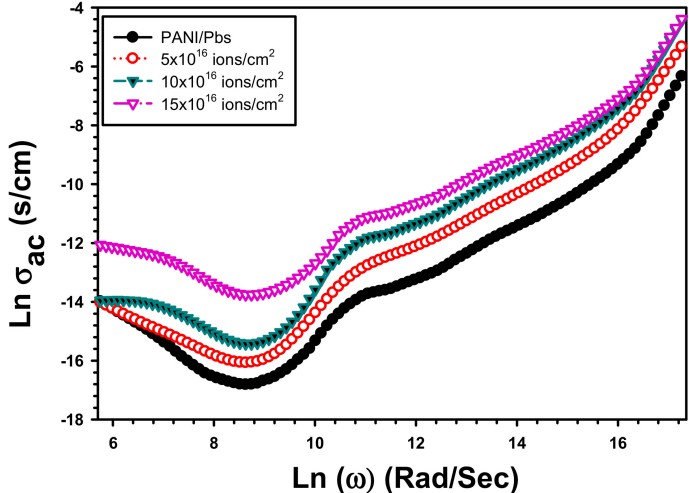

**Figure 13.** Conductivity $\sigma_{ac}$ against Ln ($\omega$) for pristine and treated PANI/PbS.

The maximum energy barrier height $W_m$ is computed using this formula [62]:

$$W_m = \frac{-4k_B T}{m} \tag{16}$$

where $T$ is temperature, $k_B$ is Boltzmann constant, and $m$ is computed from slopes of Ln ($\varepsilon''$) versus Ln ($\omega$), as observed in Figure 14, using this formula [63]:

$$\varepsilon'' = A\omega^m \tag{17}$$

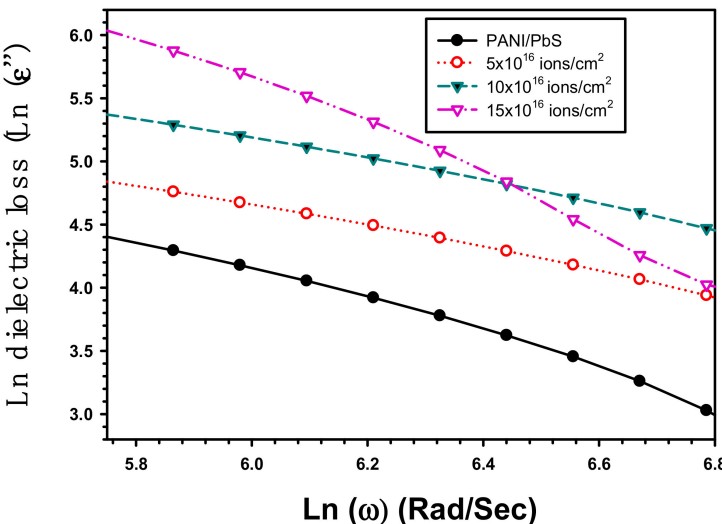

**Figure 14.** Ln ($\varepsilon''$) against Ln ($\omega$) for pristine and treated PANI/PbS samples.

The predicted $\mathbf{W_m}$ decreased from 0.43 eV for pure PANI/PbS, to 0.26 eV for $5 \times 10^{16}$ ions/cm$^2$, to 0.25 eV for $10 \times 10^{16}$ ions/cm$^2$, and to 0.23 eV by increasing the fluence to $15 \times 10^{16}$ ions/cm$^2$. This shift in $\mathbf{W_m}$ was caused by the defects created inside the polymer chains after irradiation, as recorded by the XRD measurements.

## 3. Materials and Methods

### 3.1. Synthesis

Aniline, ammonium persulfate, hydrochloric acid (HCl), lead nitrate, and sodium sulphate were purchased from Sigma-Aldrich. To get PANI, the sample was stirred, thoroughly rinsed, then dried at 55 °C for 10 h, and then PANI formed a full precipitate after 1 h. PANI was synthesized using an oxidative polymeric technique, in which 0.05 M of aniline was dissolved in 0.6 M in HCl with a magnetic stirrer for 50 min [25]. The oxidant was then abruptly added to aniline solution, resulting in aniline oxidation, and thus forming PANI, with a greenish color. Then the PANI was washed with deionized water and consequently dried for 8 h at 65 °C.

PbS was produced by ultrasonicating solutions of 0.05 M Pb(NO$_3$)$_2$ and 0.05 M Na$_2$S for 25 min. After pouring the Na$_2$S solution over the Pb(NO$_3$)$_2$ and ultrasonically treating the mixture for 1.5 h, a black precipitate formed, indicating the creation of PbS nanoparticles. The black precipitate was then heated in a microwave oven for 25 min in N$_2$ gas. Finally, the produced nanoparticles were dried at 70 °C for 20 h after being rinsed thoroughly with warm water multiple times. To create the PANI/PbS composite, we used the produced PbS nanoparticles for the deposition of PANI by oxidative polymerization of aniline. The oxidative polymerization was performed using 0.05 M aniline concentrations and 0.05 g of PbSNPs. For one hour, the solutions were ultrasonicated. After that, a magnetic stirrer was used for 5 h at a temperature of 298 K. After that, the composite was thoroughly rinsed in distilled water and dried at 80 °C for 9 h.

### 3.2. Ion Source Description

The PANI/PbS with a mean thickness of 0.05 mm was irradiated with different fluences ($5 \times 10^{16}$, $10 \times 10^{16}$, and $15 \times 10^{16}$ ions/cm$^2$) of oxygen beams using a cold cathode ion source, as previously described [26]. The ion source, as shown in Figure 15, is composed of two elements: a cylinder anode and acceleration system electrode. The plasma media is created in the cylindrical anode region, as depicted in Figure 1, and the oxygen ion beam is extracted via an extractor and then accelerated. The extracted ion beam is set at ion current density, operating pressure, and oxygen energy of 180 uA/cm$^2$, $2.0 \times 10^{-4}$ mbar, and 3.0 keV, respectively. The stopping parameters of the oxygen beam interacting with PANI/PbS were estimated using a SRIM/TRIM simulation [27]. The SRIM simulation was run taking into account the perpendicular incidence of oxygen, giving the incidence ion parameters of energy of 3 keV and oxygen ion mass of (15.995 amu). The software gives the thickness of the changed layer based on the penetration of 1000 A° and the target's density data. Analytically, the energy transmitted from one ion to an atom of the PANI/PbS is used to determine the ion distribution with recoils and rapid calculation of damage from the input data. The depth value that an ion can penetrate into the PANI/PbS lattice, as well as the mean range values can all be seen in the histogram for that energy after collisions. Moreover, it can also be used to determine the quantity of backscattered ions and vacancies.

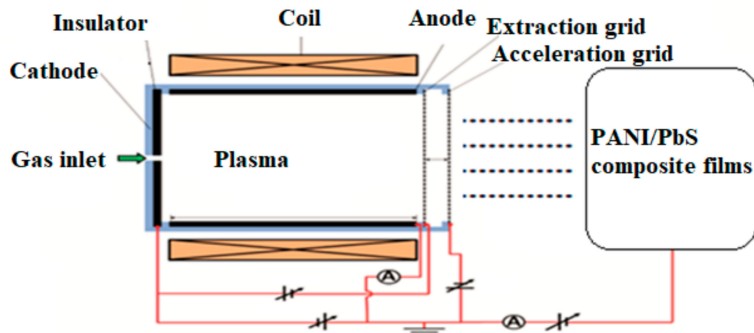

**Figure 15.** Schematic of a broad-beam ion source.

*3.3. Characterization Techniques*

XRD (Model 6000 Shimadzu) was used to analyze the structural characteristics of the pure and irradiated samples. DSC (Shimadzu, Q20, USA) was used to record the glass temperature, $T_g$, and melting temperature, $T_m$, with a heat rating of 10 °C/min. The chemical functional groups were analyzed by FTIR spectroscopy (ATI Mattson, Unicam, UK). The morphologies of the pristine and irradiated samples were examined using SEM (JEOL, Japan). The conductivity, dielectric permittivity, electrical modulus, impedance, and energy density were recorded by LCR (RS-232C, Hioki, Japan) in frequencies from $10^2$ Hz to 5 MHz.

**4. Conclusions**

In this study, PANI/PbS nanocomposites were successfully synthesized via the solution casting method and then irradiated by an oxygen beam with different fluences of $5 \times 10^{16}$, $10 \times 10^{16}$, and $15 \times 10^{16}$ ions·cm$^{-2}$. The XRD and DSC results demonstrated that the irradiation reduced the crystallite size and crystallinity degree of the composites. Both $T_g$ and $T_m$ values fell after ion irradiation; the $T_m$ decreased from 262.7 °C to 257.9 °C and the $T_g$ decreased from 130.1 °C to 120 °C after irradiation. Meanwhile, the FTIR data showed a decrease in peaks' intensities upon irradiation, suggesting the existence of chain scission in the irradiated samples. Furthermore, SEM micrographs revealed that the surface roughness is influenced by the oxygen beam irradiation. The dielectric characteristics of the pure and irradiated composites were investigated at wide range of frequencies. The irradiation caused a modification in the dielectric properties and a considerable change in the dielectric constant's coefficients. This could be attributed to chemical bonds breaking, leading to an increase in free radicals, as demonstrated by FTIR, SEM, DSC, and XRD analyses. The influence of the oxygen beam on the dielectric properties, such as energy density effectiveness and electrical modulus, of the produced samples was also investigated. Furthermore, the energy density was found to increase from $0.17 \times 10^{-3}$ J/m$^3$ for the PANI/PbS, to $2.7 \times 10^{-3}$ J/m$^3$ for the irradiated composite, indicating that the irradiated PANI/PbS film has higher impedance properties. The real modulus, **M′**, was lowered from 0.046 for pure PANI/PbS, to 0.004 for $15 \times 10^{16}$ ions.cm$^{-2}$ for irradiated PANI/PbS. These findings show that the irradiation improved the dielectric properties of PANI/PbS, meaning these films are convenient for energy devices.

**Author Contributions:** All authors shared in reviewing the whole manuscript, N.A.A. and A.A. made the methodology, R.K.S. conducted formal analysis, N.A.-H. shared the resources, N.A.A. and A.A. wrote the original manuscript, and N.A.-H. and A.M.A.H. drafted funding acquisition. All authors have read and agreed to the published version of the manuscript.

**Funding:** The authors would like to thank the Deanship of Scientific Research at Umm Al-Qura University for supporting this work by Grant Code: (22UQU4320081DSR01).

**Data Availability Statement:** The datasets generated and/or analyzed during the current study are available from the corresponding author upon reasonable request.

**Conflicts of Interest:** The authors declare no conflict of interest.

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
