# Peer review of "Surface Characterization and Electrical Properties of Low Energy Irradiated PANI/PbS Polymeric Nanocomposite Materials"

_inorganics, doi:10.3390/inorganics11020074_

Round 1

Reviewer 1 Report

1.      The Peaks of XRD patterns should be labeled.

2.      The formula (1) should be modified, there is something wrong of the cos format which was under the line.

3.      More data and evidences of PbS should be offered.

4.      The author claimed that “It is found a rise in ε' by irradiation, this because, many defects and chain scissoning produce in the composite structure.” What kind of defect? Could the author offer the data or evidence?

5.      Page 10, line 265, -2 should be superscripted.

6.      Why the surface roughness had changed after irradiation, which is the main impactor?

7.      Why the dielectric losses as well as constants coefficients are considerably altered after irradiation, much evidences should be offered.

8.      What is the concentration and size of PbS in the composite film?

Author Response

Manuscript Title: Surface characterization and electrical properties of low energy
irradiated PANI/PbS polymeric nanocomposite materials Manuscript ID: inorganics-213720

The author would like to thanks the editor and the reviewer for careful and thorough reading of this manuscript and for the thoughtful comments and constructive suggestions, which help to improve the quality of this manuscript. Our response follows (the reviewers comments are in black, while the author’s response is in highlight color).

Reviewer 1

  1. The Peaks of XRD patterns should be labeled.

Done, now the Peaks of XRD patterns was labeled (Fig.2)

  1. The formula (1) should be modified, there is something wrong of the cos format which was under the line.

Thank you for your observation, now the formula (1) is modified. (Line 100, page 3)

  1. More data and evidences of PbS should be offered.

Done, now some data and evidences of PbS is offered in the text (Line 42 to 47, page 1)

  1. The author claimed that “It is found a rise in ε' by irradiation, this because, many defects and chain scissoning produce in the composite structure.” What kind of defect? Could the author offer the data or evidence?

Thank you for your observation, now the kind and evidence of defect is discussed in the text (Line 166-170, Page 6) 

  1. Page 10, line 265, -2 should be superscripted.

Thank you for your observation, now -2 was superscripted (Line 255 page 9)

  1. Why the surface roughness had changed after irradiation, which is the main impactor?

Done, now the reason for surface roughness had changed after irradiation is discussed and mentioned in the text (Line 148-150,  Page 6) 

  1. Why the dielectric losses as well as constants coefficients are considerably altered after irradiation, many evidences should be offered.

Done, now the reason for dielectric losses as well as constants coefficients are considerably altered after irradiation with evidence is discussed and mentioned in the text (Line 188-192  Page 7)  & (Line 204-206  Page 8) 

  1. What is the concentration and size of PbS in the composite film?

Thank you for your observation, now the concentration (Line 305 page 13) and size of PbS in the composite (Line 104 page 4) is mentioned in the text.

Reviewer 2 Report

1-     The language is very bad especially grammar. It needs much polishing.

2-     Introduction should be more summarized.

3-     Figure 1: a, b, c and d must be written on images.

4-     PAGE 4 (LINE 126):  crystalline size (D) (CORRECT: crystallite)

5-     Figure 4: unit of wavenumber (cm-1)

6-     PAGE 9 (LINE 245):  In low frequency, there is an 245 exponentially of M' with increasing frequency, and at high frequency, the orientation of 246 M' is strait constant (This description is not good. The modulus seems to be wavy?! Why?)

7-     PAGE 11 (LINE 274):  conduction increases and the 274 impedance remain constant.(This a contradiction!)

8-     PAGE 12 (LINE 290):  permittivity ~ 0.885*10-12 NV/m (Is the unit true? How?)

9-     Figure 14: unit of ω is rad.

10- Conclusions must be summarized.

Author Response

Manuscript Title: Surface characterization and electrical properties of low energy
irradiated PANI/PbS polymeric nanocomposite materials Manuscript ID: inorganics-213720

The author would like to thanks the editor and the reviewer for careful and thorough reading of this manuscript and for the thoughtful comments and constructive suggestions, which help to improve the quality of this manuscript. Our response follows (the reviewers comments are in black, while the author’s response is in highlight color).

  Reviewer 2

  • The language is very bad especially grammar. It needs much polishing.

Done, now the language and especially grammar is completely edited by the English language editor to be more clearly (I attached the Certificate of Editing with the revision- and also attached the Track Changes on the supplementary file)

  • Introduction should be more summarized.

Thank you for your observation, now the introduction is modified to be more specified.

  • Figure 1: a, b, c and d must be written on images.

Thank you for your observation, now a, b, c and d is written in the images of Figure 1

  • PAGE 4 (LINE 126): crystalline size (D) (CORRECT: crystallite)

Thank you for your observation, now crystalline is modified to crystallite (Line 97-98, page 3)

  • Figure 4: unit of wavenumber (cm-1)

Thank you for your observation, now the unit of wavenumber (cm-1) is mentioned in Figure 4

  • PAGE 9 (LINE 245): In low frequency, there is an exponentially of M' with increasing frequency, and at high frequency, the orientation of M' is strait constant (This description is not good. The modulus seems to be wavy?! Why?)

Thank you for your observation, now description of the M' for low and high frequency is modified and mentioned in the text (Line 204-206 page 8)

  • PAGE 11 (LINE 274): conduction increases and the impedance remain constant. (This a contradiction!)

Thank you for your observation, now this paragraph is modified ( Line 232 page 10)

  • PAGE 12 (LINE 290): permittivity ~ 0.885*10-12 NV/m (Is the unit true? How?)

Thank you for your observation, now unit of permittivity is modified and mentioned in the text( Line 248 page 11)

9-     Figure 14: unit of ω is rad.

Thank you for your observation, now the unit of ω is modified in Fig.14)

  • Conclusions must be summarized.

Thank you for your observation, now the conclusion is completely modified to be more summarized

Round 2

Reviewer 1 Report

1.      There should be a change in the font size and font style of formula (1). All formulas should be checked.

2.      It is recommended that Table 1 and Table 2 be rearranged into three-line tables.

3.      There should be an appropriate font and font size for all labeling on the horizontal and vertical axes of all the figures.

Author Response

Manuscript Title: Surface characterization and electrical properties of low energy  irradiated PANI/PbS polymeric nanocomposite materials Manuscript ID: inorganics-213720

The author would like to thanks the editor and the reviewer for careful and thorough reading of this manuscript and for the thoughtful comments and constructive suggestions, which help to improve the quality of this manuscript. Our response follows (the reviewers comments are in black, while the author’s response is in highlight color).

Author's Reply to the Review Report (Reviewer 1)

  1. There should be a change in the font size and font style of formula (1). All formulas should be checked.

Thank you for your observation, now font size and font style of formula (1) is revised and all formula were checked.

  1. It is recommended that Table 1 and Table 2 be rearranged into three-line tables.

Thank you for your observation, now Table 1 and Table 2 were rearranged to be more  organized.

  1. There should be an appropriate font and font size for all labeling on the horizontal and vertical axes of all the figures.

Thank you for your observation, now font and font size for all labeling on the horizontal and vertical axes of all the figures were revised and appropriated.
